# Pre-Diagnostic Saliva Microbiota of School-Aged Children Who Developed Type 1 Diabetes or Inflammatory Bowel Diseases

**DOI:** 10.3390/ijms24098279

**Published:** 2023-05-05

**Authors:** Laura Räisänen, Nitin Agrawal, Binu Mathew, Sohvi Kääriäinen, Kaija-Leena Kolho, Heli Viljakainen

**Affiliations:** 1Faculty of Medicine and Health Technology (MET), Tampere University, 33100 Tampere, Finland; 2Department of Pediatrics, Tampere University Hospital, 33520 Tampere, Finland; 3Folkhälsan Research Center, 00250 Helsinki, Finland; 4Faculty of Medicine, University of Helsinki, 00014 Helsinki, Finland; 5Finnish Institute for Health and Welfare, 00271 Helsinki, Finland; 6Children’s Hospital, University of Helsinki and Helsinki University Hospital (HUS), 00290 Helsinki, Finland

**Keywords:** autoimmune disease, Crohn’s disease, microbiome, pediatric, risk factors, ulcerative colitis

## Abstract

Altered commensal microbiota composition has been associated with pediatric type 1 diabetes mellitus (T1D) and inflammatory bowel diseases (IBD), but the causal relationship is still unclear. To search for potential pre-diagnostic biomarkers for pediatric T1D or IBD, we compared microbiota in saliva samples in a nested case-control design comprising children developing T1D (n_children_ = 52) or IBD (n_children_ = 21) and controls with a similar age, sex, and residential area (n_children_ = 79). The pre-diagnostic saliva microbiota alpha- and beta-diversity of children who would develop T1D (n_samples_ = 27) or IBD (n_samples_ = 14) minimally varied from that of controls. The relative abundances of *Abiotrophia* were higher, while those of *Veillonella*, *Actinomyces*, *Megasphaera*, *Butyrivibrio*, and *Candidatus ancillula* were lower in children who would develop T1D. Within 2 years before diagnosis, the metabolic PWY-5677 pathway (converting succinate into butyrate) was lower in pre-T1D samples than in controls (*q* = 0.034). No significant pre-IBD differences were found. In conclusion, saliva microbiota diversity or composition were not successful predictors for pediatric T1D nor IBD. Intriguingly, the succinate fermentation pathway was predicted to be lowered before the onset of T1D. Thus, investigating functional pathways might provide a better approach in searching for biomarkers for autoimmune disease in the future.

## 1. Introduction

Type 1 diabetes mellitus (T1D) and inflammatory bowel diseases (IBD) are common chronic autoimmune diseases in children. For unknown reasons, their incidences among Finnish children are among the highest in the world and are continuously increasing [1]. The pathogenesis of T1D includes the autoimmune destruction of endocrine pancreatic β-cells in the islets of Langerhans by the infiltration of autoreactive T-cells [2,3], whereas the pathogenesis of IBD (including Crohn’s disease, ulcerative colitis, and IBD unclassified) is thought to be a multifactorial process involving genetic susceptibility, environmental factors, and interactions with gut microbiota. While T1D can be diagnosed using merely blood samples, IBD diagnosis in children requires invasive endoscopic procedures performed under general anesthesia. Moreover, the onset of T1D could be anticipated by the presence of pancreatic autoantibodies, while no novel IBD-specific and -sensitive autoantibody for estimating the development has been available so far [4].

Despite differences in their pathogenesis, both T1D and IBD are associated with commensal microbial alterations. Most previous studies regarding commensal microbiota and autoimmune diseases have focused on gut microbiota [5,6,7]. However, in recent times, the role of saliva microbiota in the development of pediatric autoimmune diseases has gained more attention. For instance, children with T1D have been reported to have a more abundant *Streptococcus* genus and a higher number of species in their saliva than their healthy peers [8,9], while children with IBD presented with a lower abundance of *Fusobacteria* [10]. Therefore, altered oral microbiota composition has been studied as a potential noninvasive biomarker for monitoring T1D and IBD disease activity and its response to therapy [9,11].

The detection of specific, reliable, non-invasive, and affordable predictive pre-diagnostic biomarkers for autoimmune diseases is valuable for clinical practice yet currently non-existing. Upon discovery, such biomarkers may be used not only to confirm diagnoses but also to monitor clinical attempts to alter or delay the disease course as part of the prevention program.

Despite reported associations between the altered saliva microbiota composition and autoimmune diseases, the question of whether the disrupted microbial homeostasis is a predisposing factor or a consequence of autoimmune diseases has not been answered. In this study, we had a unique opportunity to study the composition of the saliva microbiota and predict its functionality before the onset of autoimmune diseases in a nested-case control study in children. Our hypothesis was that saliva microbiota is altered already before the diagnosis of T1D and IBD during childhood.

## 2. Results

### 2.1. Background Data of Obtained Saliva Samples

This is a nested case-control study based on the Finnish Health in Teens (Fin-HIT) cohort of ~11,000 children (born 2000–2005). The saliva samples of the cohort were collected at the baseline of the Fin-HIT study (10,769 samples when the mean age of the cohort was 11 years) and at the first follow-up (5705 samples when the mean age of the cohort was 13 years). Among the children with available saliva samples, 52 children who developed T1D and 21 who developed IBD after the age of 6 years were selected, and their saliva samples were marked as cases. In addition, 79 children with compatible ages, sexes, and residential areas without registered chronic autoimmune diagnosis by the end of 2018 were closely matched from the same Fin-HIT cohort by a person outside the study group as controls. Seven children who developed T1D, two children who developed IBD, and two controls provided saliva samples at both the baseline and first follow-up. Therefore, our analysis had 59 (52 + 7) saliva samples for T1D diagnosis, 23 (21 + 2) samples for IBD diagnosis, and 81 samples (79 + 2) from controls, comprising a total of 163 saliva samples for analysis (Appendix A).

Of the saliva samples from the cases, 27 were dated before T1D diagnosis (pre-T1D samples) and 32 were dated after T1D diagnosis (post-T1D samples); 14 were dated before and 9 were dated after IBD diagnosis (pre- and post-IBD samples). To perform in-depth analysis when possible, these pre-diagnosis and post-diagnosis samples were further categorized into different timepoints based on the time of saliva collection: (1) 0–2 years before diagnosis; (2) over 2 years before diagnosis; (3) within 0–2 years after diagnosis; and (4) over 2 years after diagnosis (Table 1). The median time difference between the saliva sampling and the date of diagnosis was 1.46 (IQR 0.69–2.06) years for T1D and 1.21 (0.75–3.10) years for IBD.

Background data of the cases and controls are presented in Table 2. There were no differences in the background characteristics between the groups. The mean age at the saliva sampling was 11.9 (SD 1.6) years. The timespan between the latest antibiotic exposure and the collection of saliva samples did not differ between the cases and controls either (Table 2).

### 2.2. Alpha- and Beta-Diversity of T1D or IBD Cases vs. Controls

The differences in the saliva microbiota alpha-diversity, beta-diversity, and composition between the T1D/IBD cases and controls were characterized at different timepoints, with major focus on the samples collected before the diagnosis (Table 1). Alpha-diversity indices for T1D/IBD cases and controls were plotted according to the timing of diagnosis in Figure 1.

Within 0–2 years prior to the diagnosis, the Shannon index tended to be higher in the saliva of T1D cases than in the control samples, while the Chao1 index tended to be lower in the saliva of IBD cases than in the control samples. However, in both patient groups, significance was not achieved (crude *p* > 0.05). After diagnosis, the alpha-diversity of post-T1D samples resembled the diversity of control samples, while deviation was still present in the post-IBD samples (Figure 1), although not significant after correction with the Benjamini–Hochberg method.

Beta-diversity did not differ between pre-T1D samples (n = 27) and control samples (n = 81) (corrected *p* = 0.27) (Figure 2A). Further inspection showed that within 0–2 years before diagnosis, pre-T1D saliva microbiota (n = 20) clustered differently than the saliva microbiota of controls (n = 81) (Figure 2B) (crude *p* = 0.04). However, this difference was attenuated after correction with the Benjamini–Hochberg method (corrected *p* = 0.48). A similar finding was seen when the pre-T1D samples were compared with the T1D-specific control samples (i.e., saliva samples of controls, which were obtained at the same time as the pre-T1D samples; n = 20; crude *p* = 0.08, corrected *p* = 0.50) (Figure 2C). All corrected *p*-values were obtained using the permutational analysis of variance (PERMANOVA) test.

Regarding IBD, no differences in the clustering were seen between pre-IBD samples (n = 14) and control samples (n = 81) (*p* = 0.76) (Figure 2A). Due to a small number of samples obtained within 0–2 years before IBD diagnosis (n = 9), we abstained from further categorizing these samples based on timepoints.

Upon the comparison of the beta-diversity between pre-T1D (n = 27) and post-T1D (n = 32) samples, we found no differences (*p* = 0.96, Appendix A). In parallel to the findings regarding T1D, the beta-diversities of pre-IBD samples (n = 14) and post-IBD samples (n = 9) were also similar (*p* = 0.52, Appendix A).

#### 2.2.1. Saliva Microbiota Composition and Differential Abundance

The core saliva microbiota at the phylum level for control samples were Firmicutes (57.3%), Bacteroidetes (16.2%), Proteobacteria (12.4%), and Actinobacteria (7.6%) (Table 3). A similar phyla distribution was also seen in the saliva of T1D/IBD cases, both before and after diagnosis. Differences in the relative abundance at the genus level were compared between the saliva of T1D cases 0–2 years before diagnosis (n = 20) and the saliva of controls (n = 81). Genera such as *Abiotrophia* (from phylum Firmicutes)*, Lautropia* (Proteobacteria), and *Aggregatibacter* (Proteobacteria) were more abundant in the pre-T1D samples than in the control samples. In contrast, *Veillonella* (Firmicutes)*, Actinomyces* (Actinobacteria)*, Prevotella, Megasphaera* (Firmicutes)*, Butyrivibrio* (Firmicutes)*,* and *Candidatus ancillula* (Actinobacteria) were less abundant in the pre-T1D samples than in the control samples (Table 4).

When shrinking the saliva samples of the controls to T1D-specific control samples (n = 20), we observed that *Abiotrophia* (Firmicutes)*, Gemella* (Firmicutes)*,* and *Granulicatella* (Firmicutes) along with an uncultured genus were more abundant in the pre-T1D samples (Table 4), while *Veillonella* (Firmicutes)*, Actinomyces* (Actinobacteria)*, Candidatus saccharimonas* (Actinobacteria)*, Megasphaera* (Firmicutes)*, Butyrivibrio* (Firmicutes)*,* and *Candidatus ancillula* (Actinobacteria) were less abundant in the pre-T1D samples than in the T1D-specific control samples. None of these differences remained significant after correction with the Benjamini–Hochberg method. However, in both sets of controls, *Abiotrophia* was more abundant, and *Veillonella, Actinomyces, Megasphaera, Butyrivibrio,* and *Candidatus ancillula* were less abundant in pre-T1D samples than in these two types of control samples, further confirming these findings. In IBD, the relative abundance analysis at the genus level within 0–2 years before diagnosis was not performed due to the low number of samples (n = 9).

#### 2.2.2. Functional Prediction of Saliva Microbiota

Potential differences in saliva microbiota functions were investigated in the pre-T1D samples and in the control samples. The metabolic differences in the saliva microbiota were predicted using PICRUSt2 with the Storey correction method. One functional pathway was lowered in pre-T1D samples within 0–2 years before diagnosis when compared with the control samples: the pathway PWY-5677 fermenting succinate to butyrate (*q* = 0.034, Figure 3, Appendix A). Interestingly, this finding was no longer evident after the onset of T1D. When the corresponding comparison was conducted with the T1D-specific control samples, no pathways differed significantly.

As for IBD, a few metabolic pathways were lowered in pre-IBD samples (n = 14) when compared with the control samples (crude *p* < 0.05) (Appendix A). However, after the Storey correction method, these differences were no longer retained.

## 3. Discussion

To find a potential predictive biomarker for pediatric T1D or IBD, we investigated the saliva microbiota before the onset of these diseases using a unique nested case-control study design. Our hypothesis was that such markers could be identified in both diseases. Interestingly, we observed that the pre-diagnostic alpha-diversity in the saliva of T1D and IBD only minimally deviated from the saliva of controls. The post-diagnosis alpha-diversity in the saliva of T1D cases resembled that of the controls, but this was not the case for post-IBD samples. We speculated that starting the insulin treatment and/or depleted autoinflammatory processes may provide an explanation for this observation [12]. On the other hand, beta-diversity did not differ between pre-T1D samples and control samples nor between pre- and post-T1D samples. In general, the saliva microbiota composition of the T1D/IBD cases and the controls showed no prominent differences regarding the core taxa at the phylum level either. Nevertheless, when focusing on samples collected within two years before T1D diagnosis, a higher variation in the saliva microbiota diversity and in the relative abundance of several genera belonging to the phylum Firmicutes was observed in the pre-T1D samples. The predicted functional capacity of the saliva microbiota within this period also showed a lower proportion of the PWY-5677 pathway, converting succinate to butyrate, in comparison to the control samples.

### 3.1. Saliva Microbiota and T1D

We observed that within two years before T1D diagnosis, the Shannon index of pre-T1D samples deviated from that of the control samples. Concordantly, a previous study has pointed out a shift in gut microbiota diversity appearing approximately two years before the onset of T1D [13]. However, they reported a lowered alpha-diversity in the gut microbiota before the onset of T1D, whereas our findings suggested a numerically higher alpha-diversity in the saliva of children who would later develop T1D. In T1D, the saliva glucose concentration increases in response to the elevated blood glucose (hyperglycemia), providing more nutrition to oral bacteria [14]. This might explain the higher alpha diversity and relative abundance of several bacterial genera compared with controls before the onset of T1D. Nevertheless, children who develop T1D are usually normoglycemic years before diagnosis [15]; hence, their saliva glucose concentration is unlikely to be remarkably high. Furthermore, in comparison to the gut, saliva microbiota is considered more resilient to changes over a long period of time [16], making our results intriguing.

We showed that the abundances of four genera belonging to the Firmicutes phylum (namely, *Veillonella*, *Megasphaera, Butyrivibrio*, and *Abiotrophia*) and two genera belonging to the Actinobacteria phylum (*Actinomyces* and *Candidatus ancillula*) in the pre-T1D samples differed from the control samples within two years before T1D diagnosis. The phylum Firmicutes has been shown to play an important role in the activity of the PWY-5677 pathway, which is involved in the fermentation of succinate into a short-chain fatty acid butyrate. The role of the genera or specific species involved in butyrate metabolism is not yet well understood. A functional pathway usually involves several bacterial species, and stable pathways have been reported despite variation in microbial abundances [17]. Therefore, focusing on changes in the saliva microbial functional pathways rather than on microbial composition might be a reasonable approach to continuing the pursuit for T1D biomarkers.

As for phylum Actinobacteria, we have previously shown a link between the lower abundance of Actinobacteria and overweight/obesity [18]. In the present study, children who developed T1D had similar anthropometry to controls. Therefore, obesity is an unlikely confounding factor, and our findings on saliva microbiota within two years before the T1D diagnosis might point to an ongoing autoimmune process/es. However, this hypothesis warrants further investigation.

In the oral cavity, butyrate exhibits antimicrobial activity by inhibiting some Gram-positive oral bacterial species [19]. Thus, a lowered butyrate pathway may contribute to the higher diversity in the saliva microbiota observed in our results. In addition, decreased butyrate production has been shown to lower the activity of regulatory T-cells [19], which can trigger autoimmune responses, including T1D [20]. In summary, butyrate in the oral cavity may affect the human immune function in several ways and warrants further studies in association with pediatric T1D.

### 3.2. Saliva Microbiota and IBD

Throughout the study, alpha and beta diversity indices seemed to be lower in pre-IBD samples than in control samples. However, after correction for multiple testing, this was not a significant finding. Due to a low number of samples, we could not investigate the pre-IBD samples by timepoints, as we did for pre-T1D samples. To our knowledge, no previous studies have described the pre-diagnostic saliva microbiota composition in pediatric IBD. Furthermore, pediatric studies on saliva microbiota and IBD are scarce. A previous longitudinal study on American children showed that saliva microbiota distinguished children with IBD from healthy controls better than microbiota from other oral sites or stool [21]. In our pre-IBD samples, we did not find major differences in the saliva microbiota composition or function when compared with control samples, which was surprising.

### 3.3. Strengths and Limitations of the Study

Our core strength lies in the uniqueness of our saliva samples, especially those obtained several years before a diagnosis, when the children were still defined as healthy and free from chronic autoimmune diagnosis. By using pre-diagnostic saliva samples, we were able to compare the saliva microbiota between two groups of healthy children, of which one group would develop T1D or IBD by a mean age of 11 or 12 years. This makes even minor changes between the saliva microbiota of case and control groups noteworthy, especially from a clinical perspective. Other than age, saliva microbiota composition is also influenced by environmental and genetic factors [22,23,24]. For this reason, we carefully matched the controls by age, sex, and residential area (postal code). Thus, the cases and controls most likely originated from the same public school, and their saliva samples were collected at the same school day during the baseline recruitment. Therefore, the role of age, sex, residential area, and date of sampling as potential confounders was marginal, which was also one of the major strengths of this study. Moreover, we also had saliva samples obtained after the onset of T1D or IBD. Even when the number of samples was low, our results suggest that after T1D treatment, the saliva microbial diversity may return to a normal state, which was not seen in IBD, indicating the need for further investigations. Finally, for seven children who developed T1D between the first saliva sampling at baseline recruitment and the second saliva sampling at the first follow-up a few years later, we had both pre- and post-T1D saliva samples from the same child. This allowed us to test potential changes between the pre- and post-diagnostic saliva microbiota in the same individuals. However, the findings did not differ significantly from each other (Appendix A), and we could not perform further in-depth analysis.

Our study was performed on the data and saliva samples collected from children living in Finland, which has the highest incidence of pediatric autoimmune diseases in the world [1], therefore giving us the best chance to investigate these diseases from a pre-diagnostic point of view. We have not screened the children for genetic risk factors for autoimmune diseases. Despite high incidences in Finland, pediatric T1D and IBD are still considered relatively rare diseases, explaining the limited availability of pre-diagnostic saliva samples. For instance, in the Fin-HIT cohort of more than 11,000 children, the prevalence of T1D was 0.89%, and that of IBD was 0.25% [1]. Therefore, we could not influence the number of the available pre-diagnostic samples in this nested case-control study, and the fact that we were able to obtain as many pre-diagnostic saliva samples is exceptional. However, due to the limited sample size (especially regarding IBD), the FDR correction may be too strict and unnecessary.

We had no data on the children’s genetic background, on their innate immunity profile, nor on their family predisposition to T1D or IBD to further decrease the variability. We also lack data on when the multiple isle antibodies, i.e., seroconversion, would occur before T1D diagnosis. Therefore, we could not connect our findings to the staging of T1D. Finally, we have no guarantee that the antibiotics purchased before obtaining saliva samples were consumed. However, surveilling the antibiotic consumption of over 11,000 children for over a decade is practically unachievable. Therefore, we used antibiotic purchases as a proxy of the exposure.

Saliva microbiota composition may also be influenced by oral hygiene and health, stages of dentition, and dietary patterns—especially by the consumption of sugary products [25]. While the dietary patterns and sugary product consumption of children in the Fin-HIT cohort who developed autoimmune diseases have been reported to be similar to those of those who did not develop these diseases [26], we lacked detailed information on the children’s overall oral health as a potential confounder. Nevertheless, our earlier findings from a Fin-HIT subsample (n = 617) presented a good oral health: 66% of the children had no history of cavitated caries lesions in permanent teeth, and only minor differences in the saliva microbiota were observed between children with and without dental caries [27]. In addition, children in the current study were matched by age. Therefore, we are assuming that the cases and controls would likely be at the same stage of dentition, and the differences in their oral health may be minimal.

Finally, the prediction of the functional pathways by PICRUSt2 has approximately >80% accuracy [28], which is lower than functional profiles received through metagenomic shotgun sequencing. Moreover, metagenomics sequencing also provides a species-level identification of the microbiota, further aiding the functional profiles (currently, a limitation with the 16s rRNA sequencing). Thus, to confirm the role of saliva microbiota and the function of butyrate as a potential predictive bioindicator for the onset of T1D, studies using more advanced techniques are needed.

## 4. Materials and Methods

### 4.1. Data Sources for the Study

We used a case-control study, nested in the Finnish Health in Teens (Fin-HIT) study cohort. The Fin-HIT cohort is a nationwide school-based cohort addressing the health behaviors of Finnish children and adolescents and comprising over 11,000 children (born 2000–2005) from densely populated areas across Finland, without specific exclusion criteria. More detailed information on the recruitment process and characteristics of the cohort has been described previously [29] and can be found on the website https://www.finhit.fi/data/ (accessed on 8 February 2023). At the baseline school recruitment in 2011–2014, unstimulated saliva samples were collected from the children under the supervision of trained fieldworkers using the Oragene^®^ DNA Self-Collection Kit (DNA Genotek Inc., Ottawa, ON, Canada) [30]. Among the participants, 53% participated in the first follow-up in 2015–2016, providing repeated information on anthropometry, health behaviors, and self-collected saliva samples. The saliva samples were stored at room temperature until the analysis [31].

Using a unique personal identity code owned by every Finnish resident, children in the Fin-HIT cohort were linked to representative national registers—the Special Reimbursement Register (SRR) and the Drug Purchase Register (DPR)—maintained by the Finnish Social Insurance Institution (i.e., KELA in Finnish). The SRR contains records on patients with chronic diseases requiring continuous medication (including entry dates and physician-verified diagnoses), who are entitled to drug refunds regardless of their socioeconomic status. For this study, the data of the participants were collected from national registers until 31 December 2018. Information on children who developed T1D or IBD after the age of 6 years was obtained from the SRR using ICD-10 codes (International Classification of Diseases, 10th revision): E10 for T1D, and K50 (Crohn’s disease) or K51 (Ulcerative colitis/IBD unclassified) for IBD. Sequentially, the DPR contains data on all purchased drugs by prescriptions in Finland (including dispensation dates and pharmaceutical information). The excellent coverage of the register has been shown elsewhere [32]. Children who were diagnosed before the age of 6 years were excluded from the study to reduce the risk of genetically driven autoimmune diagnosis.

Antibiotic exposure before obtaining saliva samples was regarded as a potential confounder and was addressed. Data on antibiotic purchases before obtaining saliva samples were obtained from the DPR using ATC (Anatomical Therapeutic Chemical) codes starting with J01.

### 4.2. DNA Extraction, Amplification, and Sequencing

The DNA extraction was conducted at the Technology Center, Sequencing Unit, in the Institute for Molecular Medicine Finland (FIMM) and contained lysis and the mechanical disruption of the bacterial cells by bead-beating [33]. Amplification and sequencing libraries were prepared according to our in-house 16S PCR amplification protocol [33]. 16S primers (5′CCTACGGGNGGCWGCAG′3 and 5′GACTACHVGGGTATCTAATCC′3) were used to amplify the V3-V4 regions [34]. Amplification was performed using the Truseq (TS)-switched tail amplicons. The sequencing of PCR amplicons was performed using the 2 × 301 bp sequencing on the Illumina MiSeq PE300 instrument (Illumina, Inc., San Diego, CA, USA) at FIMM. A detailed 16S rRNA gene sequencing protocol is explained elsewhere [33].

The sequencing generated ~5 million reads from 163 samples. The mean read count per sample was 30,992 (range 15,742–66,487). The reads were categorized based on sequence similarities into 2414 Operational Taxonomic Units (OTUs), divided into 12 phyla, 18 classes, 31 orders, 65 families, and 119 genera.

### 4.3. Bioinformatics and Statistical Analysis

Sequence quality filtering processing was carried out using the CLC Genomics Workbench (Version 20.0.4) (https://digitalinsights.qiagen.com (accessed on 17 March 2021)) (RRID:SCR_011853). The SILVA 16S rRNA database (Version v132) (RRID:SCR_006423) and taxonomy was used for the alignment and classification of the sequences [35]. To ensure high-quality data for analysis, reads containing ambiguous bases, more than one mismatch in the primer sequence, fewer than 40 base pair assembly overlaps, and over 5 unaligned mismatched ends under the default parameters in CLC were removed. Assembled reads shorter than 100 bp and over 470 bp in length were excluded from the analysis. The high-quality assembled reads were aligned to the SILVA 16S rRNA database, clustered into Operational Taxonomic Units (OTUs) with a cut-off value of > 99% similarity levels among sequences, and assigned taxonomy using the SILVA bacterial taxonomy database. No samples were omitted during the analysis due to low-sequencing-depth samples (< 50% median, and lower than 10,000 reads).

To illustrate the differences within the sample, the saliva microbiota alpha-diversity indices (=Shannon, inverse Simpson, and Chao1) of T1D and IBD cases were compared to those of the controls (n = 81) using a K-Independent sample test. The beta-diversity (differences between the samples) between the saliva of T1D/IBD cases was compared first with the saliva from all controls (n = 81). Then, the saliva of T1D cases was compared to T1D-specific controls (i.e., control saliva samples, which were obtained at the same time with the pre-T1D samples, n = 20). Both times, the Principle Coordinate Analysis (PCoA) was used, with the Bray–Curtis dissimilarity index and permutational analysis of variance (PERMANOVA) using the *adonis* function from the *vegan* package (RRID:SCR_011950) with 999 permutations. The obtained *p*-values were corrected with the Benjamini–Hochberg method.

Bacterial composition was calculated at the phylum and genus levels. The association of bacterial abundance with the OTUs at a genus level was tested with General Linear Models (GLM) using a negative binomial distribution, calculating the log_2_fold changes between the bacterial taxonomies, implemented in the *DESeq2* package (version 1.34.0) (RRID:SCR_015687).

IBM SPSS Statistics 26.0 software (IBM Corp., Armonk, NY, USA) (RRID:SCR_016479) was used for analyzing background data and for comparing the alpha-diversity indices. GraphPad Prism version 9.3.1 (RRID:SCR_002798) was used for data visualization. Otherwise, the statistical analyses were conducted in R version 4.0.4 (RRID:SCR_001905) using the *Bioconductor* version 3.12.0 (RRID:SCR_006442), *Microbiome* version 1.12.0, *Vegan* version 2.5–7 (RRID:SCR_011950), and *Phyloseq* version 1.34.0 (RRID:SCR_013080) packages. Due to the limited sample size, we considered findings with *p* < 0.1 before correction as tentative.

The predicted functional pathways of saliva microbiota were compared between pre-T1D/IBD samples and control samples using Phylogenetic Investigation of Communities by the Reconstruction of Unobserved States—PICRUSt2 program version 2.0.0-b.2 [28]. The affected pathways were visualized using Statistical Analysis of Metagenomic Profiles–STAMP software version 2.1.3 (RRID:SCR_018887) [36]. The Welch test was used for comparison, and adjusted *p*-values with the Storey method were presented. The MetaCyc (RRID:SCR_007778) database (https://metacyc.org/ (accessed on 27 January 2022)) was used for pathway annotation.

## 5. Conclusions

Before the onset of T1D in children, especially within 0–2 years before diagnosis, some minor deviation of the saliva microbiota diversity and composition from the healthy population was observed, while no changes in the microbial diversity, composition, nor functional pathway were found prior to the onset of IBD. However, the lowering of the PWY-5677 butyrate-producing pathway in the saliva of healthy children within 2 years before the onset of T1D in comparison to the saliva of disease-free controls was notable.

In summary, our study showed that the saliva microbiota diversity and composition were not potential predictive bioindicators for pediatric T1D nor IBD. In contrast, reduction in the butyrate production pathway may serve as a possible metabolic bioindicator for T1D. Therefore, focusing on changes in the saliva microbial functional pathways and metagenomics could be a more reasonable approach in the future for discovering specific biomarkers for autoimmune diseases.

## Figures and Tables

**Figure 1 ijms-24-08279-f001:**
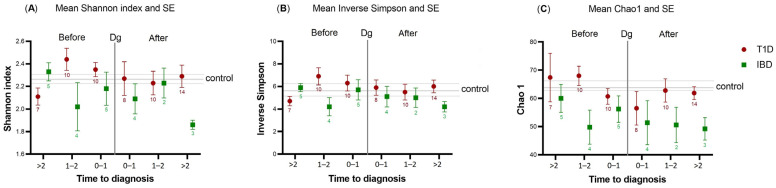
Alpha-diversity indices—(**A**) Shannon index and (**B**) Inverse Simpson—and richness index—(**C**) Chao1—plotted against the time to diagnosis (years) in children with type 1 diabetes mellitus (T1D, n = 59; red circles) or inflammatory bowel diseases (IBD, n = 23; green squares). The number of samples available in each time category before and after diagnosis is marked below the Standard Error of Mean (SE) line for T1D and IBD. The plot shows the deviations of cases against the control population (horizontal center line with the SE of the control). The vertical line marked with Dg represents the time of diagnosis. To the left of the Dg line represents pre-diagnostic samples, while to the right represents post-diagnostic samples.

**Figure 2 ijms-24-08279-f002:**
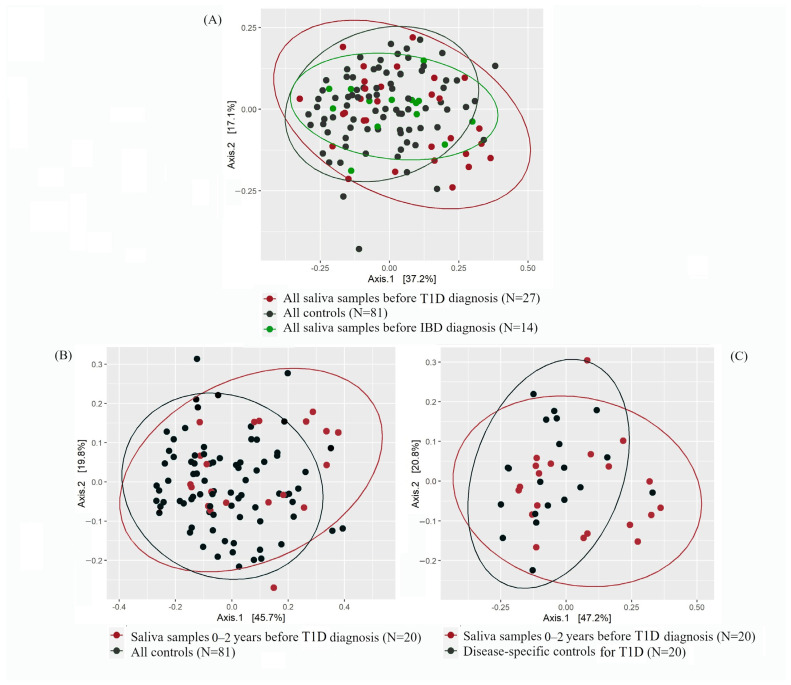
Principal coordinate analysis (PCoA) plots showing the beta-diversity based on the Bray–Curtis dissimilarity index for pre-diagnostic type 1 diabetes (T1D; in red), pre-diagnostic inflammatory bowel diseases (IBD; in green), and control samples (in black). The percentage values in Axis 1 and Axis 2 explain the largest differences in the data observed in the two coordinate axes. (**A**) All pre-diagnostic T1D and IBD samples vs. all control samples. Axis 1 and Axis 2 explain 37.2% and 17.1% of the variance, respectively. (**B**) Samples obtained 0–2 years before T1D diagnosis vs. all controls. Axis 1 and Axis 2 explain 45.7% and 19.8% of the variance, respectively. (**C**) Samples obtained 0–2 years before T1D diagnosis vs. disease-specific controls. Axis 1 and Axis 2 explain 47.2% and 20.8% of the variance, respectively.

**Figure 3 ijms-24-08279-f003:**
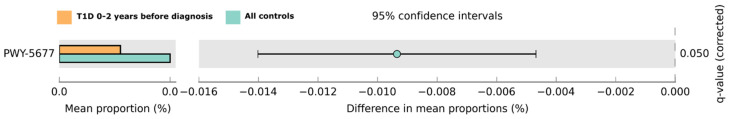
Predicted functional pathway differences in the T1D cases 0–2 years before diagnosis (n = 20; in orange) samples and all controls (n = 81; in light green). Differences in the mean proportions are shown with 95% confidence intervals. Results are based on Welch’s test; the q-value was obtained using the Storey correction method.

**Table 1 ijms-24-08279-t001:** Saliva samples based on the sampling date in relation to the date of diagnosis. Nine children had both pre- and post-diagnostic samples available (included at both timepoints).

Timing of Saliva Samples	>2 Years before Diagnosis	0–2 Years before Diagnosis	0–2 Years after Diagnosis	>2 Years after Diagnosis	Total
T1D	7	20	18	14	59
IBD	5	9	6	3	23
Controls	12	28	24	17	81

T1D = type 1 diabetes mellitus, IBD = inflammatory bowel diseases. Controls = healthy children with a similar age, sex, and residential area.

**Table 2 ijms-24-08279-t002:** Background characteristics of the participants from whom the saliva samples were obtained.

	Children Who Developed T1D (n = 52)	Children Who Developed IBD (n = 21)	Children without Autoimmune Diseases (n = 79)	*p*-Value ^a^	*p*-Value ^b^
**Age at baseline, mean ± SD**	11.0 ± 1.0	11.3 ± 0.6	11.2 ± 1.0	0.457	0.521 ^c^
**Age of diagnosis, mean ± SD**	11.2 ± 2.8	12.2 ± 2.8			
**Age at the end of December 2018, mean ± SD**	16.3 ± 1.6	16.4 ± 1.1	16.3 ± 1.5	0.994	0.662 ^c^
**Sex, n (%)**				0.672	0.606 ^d^
Boy	31 (59.6)	12 (57.1)	50 (63.3)		
Girl	21 (40.4)	9 (42.9)	29 (36.7)		
**BMI at baseline, n (%)**				0.918	0.654 ^d^
Normal	35 (67.3)	16 (76.2)	67 (84.8)		
Overweight/obese	5 (9.6)	3 (14.3)	9 (11.4)		
Missing	12 (23.1)	2 (9.5)	3 (3.8)		
**Waist-to-height ratio (cm/cm) at baseline, n (%)**					
Normal (<0.5)	35 (67.3)	18 (85.7)	69 (87.3)	0.350	0.970 ^d^
Central obesity (≥0.5)	4 (7.7)	1 (4.8)	4 (5.1)		
Missing	13 (25.0)	2 (9.5)	6 (7.6)		
**Saliva samples available for analyses at baseline/first follow-up, n**					
Baseline	41	18	59		
First follow-up	18	5	22		
Total	59	23	81		
**Antibiotic exposure (any type) prior to the collection of saliva samples, n (%)**				0.676	0.534 ^d^
No exposures	5 (8.5)	3 (13.0)	5 (6.2)		
≤3 months	2 (3.4)	1 (4.3)	5 (6.2)		
>3 months	52 (88.1)	19 (82.6)	71 (87.6)		

T1D = type 1 diabetes mellitus, IBD = inflammatory bowel diseases, BMI = Body mass index (kg/m^2^), categorization was based on International Obesity Task Force cut-offs, DMFT = Decayed, Missing, and Filled Teeth for estimating oral health. *p*-value for the comparison between: ^a^ T1D and controls; ^b^ IBD and controls. ^c^ Independent sample *t*-test. ^d^ Pearson’s chi-square.

**Table 3 ijms-24-08279-t003:** Relative abundance of the most common saliva bacterial phyla in type 1 diabetes mellitus and inflammatory bowel diseases cases and all controls. Numbers of saliva samples available are shown.

Abundance (%)	All Controls (n = 81)	Before T1D Diagnosis (n = 27)	Before IBD Diagnosis (n = 14)	After T1D Diagnosis (n = 32)	After IBD Diagnosis (n = 9)
Firmicutes	57.3	55.6	55.4	54.6	51.4
Bacteroidetes	16.2	13.6	15.3	14.5	19.1
Proteobacteria	12.4	16.7	15.2	17.4	16.4
Actinobacteria	7.6	8.0	9.3	7.5	8.7
Fusobacteria	3.1	3.1	2.4	3.2	1.8
Patescibacteria	2.6	2.3	1.9	2.2	2.1
Other	0.7	0.6	0.5	0.6	0.6
TOTAL	100	99.9	99.9	100	100

T1D = type 1 diabetes mellitus, IBD = inflammatory bowel diseases.

**Table 4 ijms-24-08279-t004:** The ten most differentially abundant genera of bacteria in the pre-diagnostic saliva samples of children later diagnosed with type 1 diabetes mellitus (T1D). The genera are listed based on the descending log2fold change value. A positive log2fold value means a higher abundance and negative value means a lower abundance in T1D compared to control samples.

0–2 Years before Diagnosis T1D Samples (n = 20) vs. All Control Samples (n = 81)					0–2 Years before Diagnosis T1D Samples (n = 20) vs. Disease-Specific Controls (n = 20)				
Genera	Base Mean	Log2 Fold Change	*p*-Value		Genera	Base Mean	Log2 Fold Change	*p*-Value	
			Crude Corrected *					Crude Corrected *	
*Abiotrophia*	4.06	1.27	0.02	0.42	*uncultured-02*	10.1	1.86	0.02	0.45
*Lautropia*	20.0	0.94	0.01	0.38	*Abiotrophia*	5.88	1.55	0.02	0.45
*Aggregatibacter*	38.9	0.90	0.03	0.56	*Gemella*	128	1.20	0.002	0.30
*Veillonella*	3133	−0.64	0.003	0.22	*Granulicatella*	59.1	0.89	0.01	0.45
*Actinomyces*	48.4	−0.71	0.008	0.24	*Veillonella*	2819	−0.51	0.04	0.53
*Prevotella 7*	785	−0.72	0.007	0.24	*Actinomyces*	47.7	−0.75	0.03	0.45
*Prevotella 6*	39.5	−0.95	0.01	0.30	*Candidatus Saccharimonas*	35.3	−0.90	0.02	0.45
*Megasphaera*	88.0	−1.28	0.001	0.21	*Megasphaera*	70.8	−1.19	0.02	0.45
*Butyrivibrio 2*	6.32	−1.55	0.005	0.24	*Butyrivibrio 2*	5.52	−1.58	0.02	0.45
*Candidatus Ancillula*	1.38	−1.88	0.02	0.42	*Candidatus Ancillula*	1.45	−2.37	0.04	0.53

* *p*-value corrected with the Benjamin–Hochberg method.

## Data Availability

The datasets generated and/or analyzed during the current study are available in the European Genome-phenome Archive (EGA) repository, wwwdev.ebi.ac.uk/ega/studies/EGAS00001006949 (accessed on 8 February 2023), and full access to the repository could be arranged on request from the corresponding author. The data are not publicly available due to restricted consent from the participants.

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
