# Peer review of "Pre-Diagnostic Saliva Microbiota of School-Aged Children Who Developed Type 1 Diabetes or Inflammatory Bowel Diseases"

_ijms, 2023, doi:10.3390/ijms24098279_

Round 1

Reviewer 1 Report

This interesting study examined the saliva microbiota before the onset of Type 1 diabetes and Inflammatory Bowel Disease in Finnish children using a unique nested case-control study design. While the results are clearly presented and represent an essential contribution to the limited knowledge of the subject, there is a lack of in-depth analysis and interpretation of the data. I suggest including a more comprehensive discussion of the findings, including possible explanations for the observed results. In addition, the authors should carefully justify the low sample size used in the study based on previous studies.  

Points to consider in subsequent versions:

1.    The abbreviation DM (for Type 1 diabetes) is misleading as it is often used for  Diabetes mellitus in general. I suggest using T1D as the acronym for Type 1 diabetes.  

2.     The authors conclude that saliva microbiota diversity or composition is not a potential predictive bioindicator for pediatric T1D or IBD. However, this study is restricted to children from Finland and is therefore not necessarily representative of genetically ‘normal’ children in other regions of the world. I wondered if the authors could consider this as another potential study limitation.  

3.     A remarkable result from the present study was the reduction of the butyrate production pathway in the saliva of healthy children within 2 years before the onset of T1D in the present study. In contrast, previous studies have shown that although there are significant shifts in taxonomic composition over time, the relative abundance of metabolic pathways within individuals in children who progress to T1D remains remarkably constant across time (Kostic AD, et al. The dynamics of the human infant gut microbiome in development and in progression toward type 1 diabetes. Cell Host Microbe 2015, 17, 260-73. https://doi.org/10.1016/j.chom.2015.01.001.PMID: 25662751). I wish the authors were more analytical on the possible reasons for the changes in the saliva microbial functional pathways observed in the present study.

4.    The authors mention that the variation in saliva microbiota diversity within two years before the T1D diagnosis might point to an ongoing autoimmune process/es. How does this relate to the staging of pre-type 1 diabetes? Could the features of the microbial community distinguish the T1D disease state? This should be more fully discussed.

5.     The return of the saliva microbial diversity to normal after T1D treatment is fascinating and seems to indicate that insulin treatment and glycemic control restore saliva microbiota. Given that dysglycemia impacts microbiota, I encourage the authors to consider that the disrupted composition of the saliva microbiota before the T1D onset could be a consequence of impaired glucose tolerance already observed in Stage 2 T1D.This will aid in answering the question of whether disrupted microbial homeostasis is a predisposing factor or a consequence of autoimmune diseases. 

Reviewer 3 Report

This is a very interesting hypothesis to be tested in a really great and rare cohort of human subjects. The objectives are clearly stated from the beginning and the data presentation flow is articulate.

The authors have made significant efforts to analyze saliva microbiota in human samples and sufficiently present all the necessary data obtained from these studies.

However, the one and only major pitfall of the study is the fact that there were no statistically significant differences in any of the parameters tested. Even the pathway enrichment analysis is based on trends and not on statistically significant differences. different That means that the original hypothesis is rejected.

That been said, this a “negative results paper”, which although is not impactful in terms of originality and ground-braking results, it would be of interest to the scientific community. Especially to people who work in the saliva biomarkers and oral microbiome fields.

Round 2

Reviewer 2 Report

Thanks for the responses to my previous list of comments. The authors have taken all comments into account including revisions and clarifications. The low participation rate remains the major issue with the study, but the work of getting a group together should be acknowledged. As the authors have commented on this clearly now, it is up to the reader to draw conclusions if the results are to be considered representative for the study groups or not.

There are, however, still editorial corrections to be done. To mention some.

-       In the abstract: the expression “corrected q” is not right. Either corrected p or q.

-       The new Supplementary Figure 1 is not possible to read. I suggest a larger font and a more contrasting color on the text against a lighter background. Further, the document with supplementary information was not updated:

-       For Table 2 it is confusing that (a) and (b) are not in superscript to match the footnotes and also that there is no indication

-       I suggest using “corrected” for p-values consistently to avoid confusion with model adjustment unless that is what you mean (e.g., Table 4).
